

# Lack of a genetic cline and temporal genetic stability in an introduced barnacle along the Pacific coast of Japan

Takefumi Yorisue[1,2]

[1] Institute of Natural and Environmental Sciences, University of Hyogo, Sanda, Hyogo, Japan
[2] Museum of Nature and Human Activities, Hyogo, Sanda, Hyogo, Japan

## ABSTRACT

**Background:** Large numbers of exotic marine species have been introduced worldwide. Monitoring of introduced species is important to reveal mechanisms underlying their establishment and expansion. *Balanus glandula* is a common intertidal barnacle native to the northeastern Pacific. However, this species has been introduced to Japan, South America, South Africa, and Europe. While a latitudinal genetic cline is well known in its native range, it is unclear whether such a genetic cline occurs in introduced areas. Twenty years have passed since it was first identified in Japan and its distribution now ranges from temperate to subarctic regions.
**Methods:** In the present study, we examined genotypes of cytochrome oxidase subunit I (COI) of mitochondrial (mt)-DNA and elongation factor 1a (EF1) across the distribution of *B. glandula* in Japan at high and mid intertidal zones.
**Results:** At all sampling sites, native northern genotypes are abundant and I did not detect significant effects of latitude, tide levels, or their interaction on genotypic frequencies. Further, I did not detect any change of genotype composition between data collected during a study in 2004 and samples in the present study collected in 2019. Data from the present study offer an important baseline for future monitoring of this species and supply valuable insights into the mechanisms of establishment and expansion of introduced marine species generally.

## INTRODUCTION

Marine organisms have been introduced beyond their native ranges by humans both intentionally and unintentionally at increasing rates over the last 200 years (*Ruiz et al., 2009*; *Galil, Clark & Carlton, 2011*). Such introduced species are recognized as major threats to coastal ecosystems (*Havel et al., 2015*, *Papacostas et al., 2017*). In fact, coastal habitats are among the most heavily invaded systems on earth (*Ruiz, Hines & Grosholz, 1999*; *Grosholz, 2002*). Therefore, understanding processes of introduction and establishment is a central goal of invasion biology.

The barnacle, *Balanus glandula*, is a dominant species in the intertidal zone of the Northeastern Pacific coast from Alaska to California. It shows a strong latitudinal genetic cline both in mitochondrial COI and nuclear EF1 genes (*Sotka et al., 2004*; *Wares &*

Corresponding author
Takefumi Yorisue,
yorisue@gmail.com

*Cunningham, 2005*). Previous studies have implicated selection and gene flow limitations in maintaining the strong genetic cline (*Sotka et al., 2004*; *Barshis et al., 2011*; *Wares & Skoczen, 2019*; *Wares, Strand & Sotka, 2021*). In marine systems, thermal selection creates intraspecific genetic structures (*Flight & Rand, 2012*; *Teske et al., 2019*; *Nunez et al., 2020*, *2021*). By means of hull fouling and/or ballast water, *B. glandula* was introduced to the Atlantic coasts of Argentina (*Rico & Gappa, 2006*), South Africa (*Simon-Blecher, Granevitze & Achituv, 2008*), and Europe (*Kerckhof, De Mesel & Degraer, 2018*), and the Pacific coast of Japan (*Kado, 2003*). In Japan, it was first detected in northern Honshu and southern Hokkaido in 2000 (*Kado, 2003*). Then, it expanded to eastern Hokkaido in the 2000s (*Alam et al., 2014*). *Geller et al. (2008)* analyzed COI and nuclear EF1 sequences of introduced *B. glandula* population in Japan and Argentina, which suggested possible involvement of thermal selection in the invasion process. This study analyzed each Japanese *B. glandula* population in northern Honshu and southern Hokkaido, and concluded that the origin(s) of the Japanese populations was somewhere between Puget Sound and Alaska (*Geller et al., 2008*). However, it is unclear whether a genetic cline exists in introduced regions, because only limited numbers of populations were analyzed (*Geller et al., 2008*). Body temperatures of *B. glandula* are closely tied to air temperatures during low tides (*Harley & Lopez, 2003*). In the distribution of *B. glandula* in Japan, there is a latitudinal gradient in air temperature (Fig. S1). At the southern edge of the Japanese *B. glandula* distribution, air temperatures in summer are comparable to those of California, its native southern region (*Geller et al., 2008*; Fig. S1). Therefore, the contribution of native southern populations may be higher in the northern Honshu populations than was suggested in the previous study (*Geller et al., 2008*) and/or genetic composition may have gradually changed if the thermal environment contributes to maintain the genetic cline in the native range. In terms of the thermal environment, it is important to consider the effect of tide levels on genetic variation (*Flight & Rand, 2012*; *Nunez et al., 2020*, *2021*). Generally higher tide levels result in stronger temperature and desiccation stresses for intertidal species when exposed at low tide. About two decades have passed since the first discovery of *B. glandula* in Japan, and I have tested genetic clines and temporal changes.

In the present study, I analyzed COI and EF1 sequences of *B. glandula* from two tidal levels across its range in Japan. Then I evaluated effects of latitude and tide levels on genotypic frequencies. Additionally, I evaluated a possible temporal change of genotypic frequencies by analyzing data obtained in 2004 and in the present study 2019.

## MATERIALS AND METHODS

### Sample collection

*B. glandula* samples were collected from nine sites in northern Honshu and Hokkaido, Japan in 2019. At each site, samples were collected from high- and mid-intertidal zones. At Ofunato and Muroran, *B. glandula* occurred only in the high intertidal zone. All samples were fixed and stored in 99.5% EtOH. Details of samples and sampling sites are given in Table 1.

**Table 1** Details of *Balanus glandula* samples used in this study.

| Site | Region | Coord | COI (High) | COI (Mid) | EF1 (High) | EF1 (Mid) |
|---|---|---|---|---|---|---|
| Nemuro (NM) | Hokkaido | 43.17775, 145.51194 | 16 (12, 4, 0) | 15 (12, 2, 1) | 28 (0, 28, 0) | 26 (2, 24, 0) |
| Akkeshi (AK) | Hokkaido | 42.986551, 144.890837 | 16 (12, 4, 0) | 15 (9, 3, 3) | 28 (0, 28, 0) | 22 (0, 22, 0) |
| Tomakomai (TM) | Hokkaido | 42.62811, 141.61529 | 16 (12, 2, 2) | 16 (13, 2, 1) | 28 (0, 28, 0) | 30 (2, 28, 0) |
| Muroran (MR) | Hokkaido | 42.356210, 141.052457 | 20 (16, 3, 1) | 0* | 34 (0, 34, 0) | 0* |
| Erimo (ER) | Hokkaido | 41.931319, 143.245847 | 16 (14, 2, 0) | 16 (12, 3, 1) | 20 (0, 20, 0) | 24 (0, 24, 0) |
| Erimo (ER) 2004** | Hokkaido | 42, 143 | 34 (21, 10, 3) | | 48 (4, 44, 0) | |
| Hachinohe (HC) | Honshu | 40.53871, 141.55843 | 16 (9, 4, 3) | 16 (13, 1, 2) | 26 (0, 24, 2) | 20 (0, 20, 0) |
| Miyako (MY) | Honshu | 39.611733, 141.963562 | 16 (15, 0, 1) | 16 (14, 2, 0) | 29 (0, 28, 1) | 26 (0, 26, 0) |
| Kamaishi (KM) | Honshu | 39.248650, 141.898030 | 15 (12, 2, 1) | 16 (13, 2, 1) | 28 (0, 28, 0) | 32 (0, 32, 0) |
| Ofunato (OF) | Honshu | 39.019545, 141.760030 | 27 (22, 5, 0) | 0* | 52 (0, 52, 0) | 0* |
| Ofunato (OF) 2004** | Honshu | 39, 141 | 17 (13, 4, 0) | | 24 (0, 24, 0) | |

Notes:
Coord, geological coordination; COI (High), number of cytochrome oxidase subunit I (COI) sequences determined from the high tide zone; COI (Mid), number of COI sequences determined from the mid tide zone; COI (High), number of elongation factor 1a (EF1) lineages determined from the high tide zone; number of EF1 lineages determined from the mid tide zone. Numbers in parentheses denote the number of haplotype groups A, B, and C, respectively.

\* *B. glandula* occured only at high tide zone.

\*\* Data from *Geller et al. (2008)*.

## DNA extraction, PCR, and sequencing

DNA was extracted from adductor muscles of each specimen using a QuickGene DNA tissue kit (KURABO) following the manufacturer's instructions. PCR of the cytochrome oxidase subunit I (COI) region of mitochondrial (mt)-DNA was performed using the universal primers LCO1490 and HCO2198 (*Folmer et al., 1994*). PCR for the elongation factor 1a (EF1) region of nuclear DNA was performed using the primers Ef1_for and Ef1_rev (*Sotka et al., 2004*). For PCR of both DNA regions, BLEND Taq-plus-(Toyobo) was used with conditions as follows: COI, an initial denaturation step at 94 °C for 2 min; 35 cycles each at 94 °C for 30 s, annealing at 54 °C for 30 s, and extension at 72 °C for 30 s, with a final extension step at 72 °C for 5 min. EF1, an initial denaturation step at 94 °C for 2 min; 35 cycles each at 94 °C for 30 s, annealing at 64 °C for 30 s, and extension at 72 °C for 20 s, with a final extension at 72 °C for 5 min. PCR products were purified using AMPure XP beads (Beckman Coulter, Brea, CA, USA) and cycle sequencing was performed using a BigDye™ Terminator v3.1 Cycle Sequencing Kit (Applied Biosystems, Waltham, MA, USA). Sequence data were obtained in forward and reverse directions by direct sequencing using a genetic analyzer (ABI 3500XL; Applied Biosystems, Waltham, MA, USA). For COI, sequences were determined and all sequences were classified into haplotype groups A to C based on criteria from previous studies (*Sotka et al., 2004*; *Geller et al., 2008*). For EF1, which is diploid, low sequence data were checked manually and classified into haplotype groups A to C based on criteria from previous studies (*Sotka et al., 2004*; *Geller et al., 2008*).

## Sequence analyses

I applied a binomial regression model, using a series of models incorporating latitude, tide level, and interactions between these factors to predict the frequency of COI and EF1 lineages (full model: glm (haplotype lineages ~ latitude * tide level, family="binomial")).

In these models, haplotype groups of COI-A and EF1-B were classified as the northern lineage, and other groups were classified as the southern lineage. The best-fit model was the one with the highest Akaike Information Criterion (AIC) weighting. These analyses were conducted in the R statistical environment (ver. 4.0.4, *R Core Team, 2021*). I further evaluated genetic structure based on COI and EF1 sequences between Hokkaido and Honshu with Analysis of Molecular Variance (AMOVA) using Arlequin 3.5 (*Excoffier & Lischer, 2010*). AMOVA results were tested for significance across 10, 100 permutations. Haplotype and nucleotide diversities of each location were calculated using Arlequin 3.5 (*Excoffier & Lischer, 2010*). As tide levels do not influence genotypic frequencies (see results), data from high and mid intertidal zones were pooled for AMOVA and calculation of genetic diversities. Fisher's exact test was performed to compare the composition of COI and EF1 haplotype groups A-C between 2004 and 2019 both at Erimo and Ofunato. In addition, $F$st values with pairwise differences between 2004 and 2019 were calculated based on both COI and EF1 sequences with using Arlequin 3.5 (*Excoffier & Lischer, 2010*). For the $F$st calculation, data from all Japanese samples were pooled. $F$st results were tested for significance across 10, 100 permutations.

## RESULTS

I determined 607 bp of partial COI sequence from 268 *B. glandula* specimens and 292 bp of 540 bp of EF1sequences. For EF1, 453 sequences were classified into haplotype groups A–C. At all sampling sites in both the high- and mid-intertidal zones, haplotype groups COI-A and EF1-B were most abundant (Fig. 1). These haplotype groups are also most abundant in Alaska and Puget Sound, which are located in the northern part of the native range (*Sotka et al., 2004*; *Geller et al., 2008*).

From the generated models for COI, AIC weighting indicated the best model to be the null model (AIC weight, 0.392). For EF1, AIC weighting indicated the best model to be all interactions (AIC weight, 0.436). In this model, however, no significant associations were detected with latitude ($P = 0.384$) or tide level ($P = 0.145$), or interactions between latitude and tide levels ($P = 0.136$) with EF1 lineages. AMOVA detected no significant population genetic structure between the Honshu and Hokkaido regions, and among populations within these regions (Table 2). No significant differences were detected in the proportion of haplotype groups of COI (Fisher's exact test, $P = 0.238$ for Erimo; $P = 0.716$ for Ofunato) or EF1 (Fisher's exact test, $P = 0.118$ for Erimo; $P = 1.000$ for Ofunato) between 2004 and 2019. In addition, $F$st values were not significantly different from zero between 2004 and 2019 in COI or EF1 (Table 3). Genetic diversity showed no clear relationships with either location or year (Figs. S2, S3).

## DISCUSSION

The main vectors for non-native marine species are ballast water and ship fouling (*Bailey et al., 2020*). Because global maritime traffic and accompanying invasion risk will continue to increase for at least several decades (*Sardain, Sardain & Leung, 2019*), clarifying and monitoring pathways of non-native species are fundamental to manage marine habitats. *Balanus glandula*, which is native to the North American Pacific coast from Alaska to

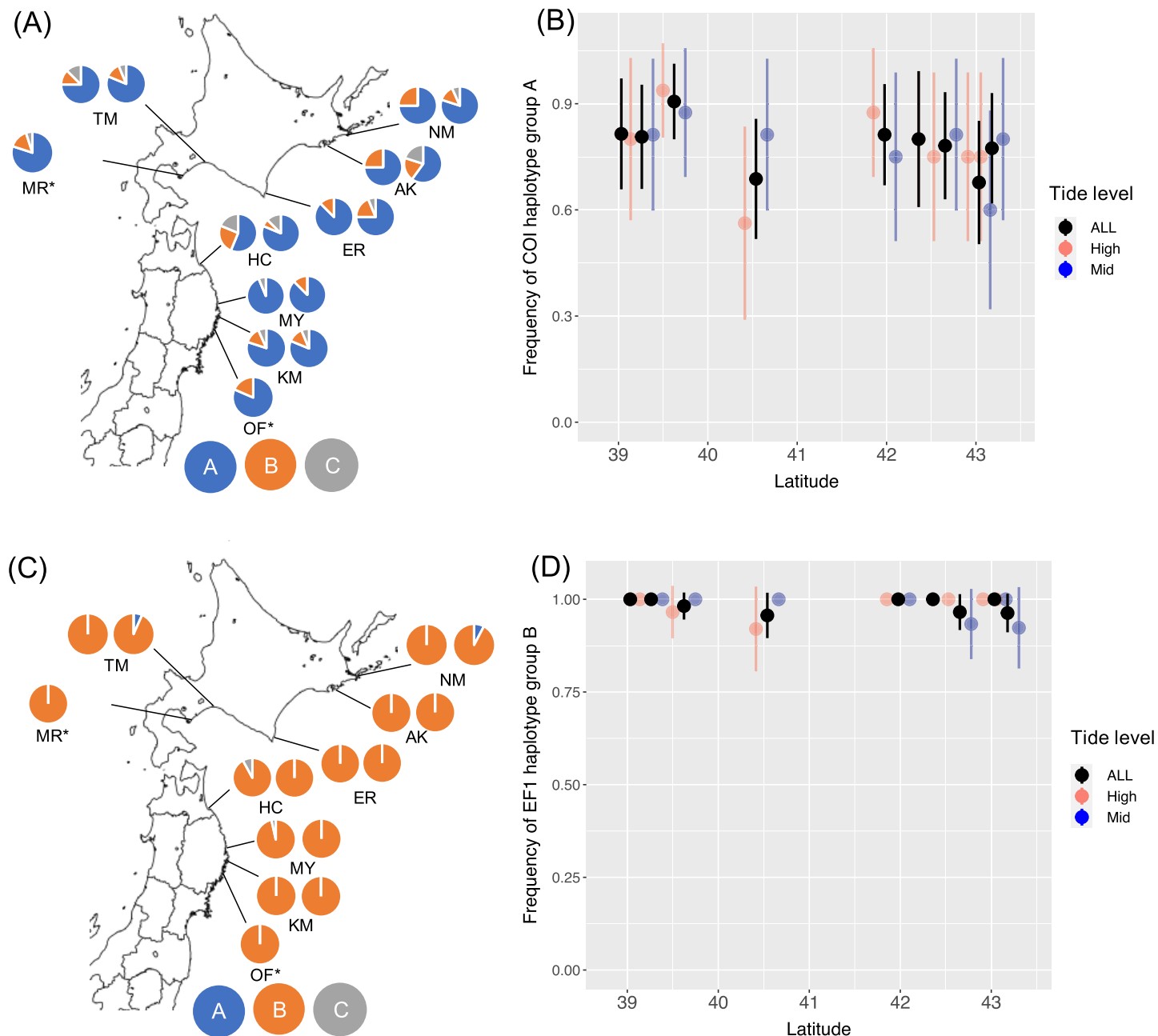

Figure 1 **Frequencies of haplotype groups in *Balanus glandula*.** (A) A map showing frequencies of COI haplotype groups with the left and right sides showing high- and mid-tide levels, respectively. (B) Frequency of COI haplotype group A (northern lineage) with latitude. (C) A map showing frequencies of EF1 haplotype groups with left and right sides showing high- and mid-tide levels, respectively. (D) Frequency of EF1 haplotype group B (northern lineage) across latitude. Bars indicate 95% confidence intervals. *B. glandula* occurred only at the high-tide zone in MR and OF.

California, was first detected in Japan in 2000 (*Kado, 2003*). Based on a previous genetic study, the source of Japanese *B. glandula* was between Alaska and Puget Sound, Canada (*Geller et al., 2008*). In the native range, a latitudinal genetic cline, which is maintained by selection (*Wares, Strand & Sotka, 2021*), has been reported (*Sotka et al., 2004*; *Wares &*

**Table 2 Summary of analysis of molecular variance (AMOVA).**

| DNA marker | Source | Df | SS | Var | % | F | P-value |
|---|---|---|---|---|---|---|---|
| COI | Among regions | 1 | 4.752 | 0.016 | 0.54 | −0.005 | 0.592 |
| | Among pops | 7 | 18.167 | −0.015 | −0.50 | 0.000 | 0.784 |
| | Within pops | 259 | 787.962 | 3.042 | 99.96 | 0.005 | 0.015 |
| | Total | 267 | 810.881 | 3.044 | | | |
| EF1 | Among regions | 1 | 0.245 | 0.001 | 0.70 | −0.006 | 0.627 |
| | Among pops | 7 | 0.436 | −0.001 | −0.61 | 0.001 | 0.827 |
| | Within pops | 531 | 51.997 | 0.098 | 99.91 | 0.006 | 0.198 |
| | Total | 539 | 52.678 | 0.098 | | | |

Note:
Df, degrees of freedom; SS, sum of squares deviations; Var, variance components; % denotes the percentage of total variance contributed by each component, $F$, fixation index.

**Table 3 $F$st and $F$st P values between time series Japanese *Balanus glandula* populations that collected in 2004 (*Geller et al., 2008*) and 2019 (present study).**

| DNA marker | Fst | P-value |
|---|---|---|
| COI | 0.008 | 0.100 |
| EF1 | 0.000 | 0.999 |

*Skoczen, 2019*). However, it is unknown whether such a cline occurs in the introduced range, because only a limited number of populations were analyzed in a previous genetic study (*Geller et al., 2008*). By resampling genotypic data across a latitudinal range at two tide levels in Japan, I evaluated effects of latitude and tide level on genotypic composition. Additionally, 15 years have passed since field sampling for the genotypic study of *Geller et al. (2008)*, so I had an opportunity to evaluate temporal changes in genotypic composition.

Binomial regression of both mitochondrial COI and nuclear EF1 lineages against latitude and tide level suggests that these factors do not affect genotypic composition. Further, AMOVA based on COI and EF1 sequences did not show genetic structure between Honshu and Hokkaido, and among populations in these regions. In other words, results of the present study did not show a genetic cline in COI and EF1 markers in *B. glandula* along the Pacific coast of Japan, whereas *Sotka et al. (2004)* detected a clear latitudinal genetic cline in the native range. While summer air temperatures in the Japanese range of *B. glandula* are comparable to those of California, the winter air temperatures are much lower than in the eastern Pacific where a genetic shift occurs in the native range (Fig. S1) (*Sotka et al., 2004*; *Geller et al., 2008*). Cold winter temperature in Japan may have caused cold temperature selection resulting in an abundance of native northern genotypes in Japan. Future studies such as genome-wide analyses with accurate environmental data are needed to evaluate the occurrence of any genetic cline that is related to environmental adaptation in this barnacle in introduced regions. In terms of thermal environments in intertidal zones, caution must be exercised because temperature and desiccation stresses can vary among latitudes due to the difference in timing/dynamics

of tide cycles (*Helmuth et al., 2006*). Reasonably priced, specialized temperature loggers can provide accurate body temperature data of intertidal barnacles (*Chan et al., 2016*).

While northern genotypes have increased in the native range through time (*Wares & Skoczen, 2019*), I did not find any evidence of a change in genotypic composition in introduced populations in Japan. *Kado (2003)* suggested independent introduction of this species in three ports (Ofunato, Hachinohe, and Tomakomai) *via* timber importation. In the present study, my data also indicated that the Alaska/Puget Sound population(s) is the main source of Japanese *B. glandula*, as suggested by *Geller et al. (2008)*, throughout its distributional range in Japan.

## CONCLUSIONS

I detected no structuring of genetic variation in the introduced barnacle, *B. glandula*, either in regard to latitude or tide level in COI and EF1, and it is inferred to be a proxy for the global population structure of this species. Genotypic composition of these genetic markers has remained stable since 2004 in Japan. Data from the present study offer an important baseline for future monitoring of this species, which provide insight into mechanisms of establishment and expansion of introduced species.

### Funding
This project was supported by grants-in-aid of the MIKIMOTO Fund for Marine Ecology and JSPS KAKENHI Grant number 20K15576. The funders had no role in study design, data collection and analysis, decision to publish, or preparation of the manuscript.

### Grant Disclosures
The following grant information was disclosed by the authors:
MIKIMOTO Fund for Marine Ecology and JSPS KAKENHI: 20K15576.

### Competing Interests
The authors declare that they have no competing interests.

### Author Contributions
- Takefumi Yorisue conceived and designed the experiments, performed the experiments, analyzed the data, prepared figures and/or tables, authored or reviewed drafts of the article, and approved the final draft.

### Field Study Permissions
The following information was supplied relating to field study approvals (*i.e.*, approving body and any reference numbers):
No permission was needed to collect *Balanus glandula* at any sampling site in this study.

### Data Availability
The COI and EF1 sequences are available at DDBJ/EMBL/GenBank: LC708364–LC709171.

## Supplemental Information

Supplemental information for this article can be found online at http://dx.doi.org/10.7717/peerj.14073#supplemental-information.

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
