# Peer review of "Lack of a genetic cline and temporal genetic stability in an introduced barnacle along the Pacific coast of Japan"

_PeerJ, doi:10.7717/peerj.14073_

## Round 0.1 · original submission · Major Revisions

Both reviewers have provided some very detailed comments and suggestions for additional analyses/edits that I think would improve the paper.

Reviewer 1 ·

Excellent Review

This review has been rated excellent by staff (in the top 15% of reviews)
EDITOR COMMENT
I want to thank the reviewer for providing a very detailed and thorough review. Specifically, I think the suggestions of validating the findings to evaluate statistical power, or testing the hypothesis about the ancestral population are do-able without additional data collection.

Basic reporting

In the paper, the author seeks to address an interesting question in ecological genetics: the patterns of population structure of a marine invasive invertebrate in a novel habitat (B. glandula invaded the habitat 20 y ago). The author describes the study hypotheses and findings using succinct and professional English. The background of the study is properly referenced with citations. However, I find that figure 1 (see general comments) needs to be redesigned to improve clarity. Also, an explicit data sharing statement is missing.

How to improve these shortcomings?

Data deposition: The data used for this study is properly described as part of the methods, but there is no data deposition/availability statement per se. I recommend moving Lines 100-101 to its own section of data availability. The DNA sequences provided in the supplementary materials look fine. The NCBI/Genbank accession numbers provided have not been released to the public so I cannot check the submissions, but I imagine that the Genbank records will be released to the public when their paper is published.

Field Studies Permits: No mention of field permits can be found in the paper. If field permits were used, they should be referenced. If the study was conducted on public lands and no public permits are needed, or if the country uses a de facto free roaming and/or free foraging law (e.g., Sweden’s Allemansrätt) it must be clearly stated as an ethics statement.

Experimental design

The study constitutes primary research and has a well-defined question as well as methods. The author seeks to test questions of population structure of B. glandula in Japan. Moreover, the author describes a knowledge gap, vis-à-vis understanding population dynamics of recent invaders, which anchors the paper’s goals. In terms of fieldwork, the study has a very detailed and granular sample range across 9 sites in northern Japan, sampled across two tidal levels. The methods are described adequately. Unfortunately, I remain unconvinced that the experimental design has sufficient power to accomplish its goals -- please see the “validity of findings section” for further explanation as well as suggestions for improvement.

Validity of the findings

The paper makes two fundamental claims about population structure of B. glandula. The first claim is introduced in line 152 and argues that there is a lack of spatial or temporal structure in the species in Japan between two sites. The second, introduced in line 154, argues that there is no temperature selection acting on the species. I argue that the data lacks the statistical power to answer these questions such that these claims are fundamentally unsubstantiated. See my reasoning below:

The issue of spatial/temporal structure (claim in line 152) is not related to any idiosyncratic property of barnacles, but rather with the first principles of population genetics needed to address the question. In 2006, Patterson, Price, and Reich published a paper (doi:10.1371/journal.pgen.0020190) where they showed that a dataset’s capacity to detect genetic structure in a population is a predictable property. Accordingly, they derived a power formula to assess the number of markers and individuals needed to detect structure as FST = 1/sqrt(n*m) … see page 2083 of their paper. Where n and m are individuals and markers respectively. While this formula was originally designed for PCA, it has general implications for genetic data. The formula argues that for each level of true “FST” (i.e., pop. structure) there is a minimum number of (unlinked markers)*(individuals) needed to detect it, and that below it, the data has no power to make any assertions about structure. Given that the author uses two linked markers and ~15 individuals per population, I am doubtful that the dataset has any power to substantiate the claim. For example, much of the demographic history of a different barnacle, S. balanoides, was done using a couple of mtDNA markers (D-loop and COX I) in the early 2000s. Using these markers, one could barely detect differences at the spatial level of entire continents thus leading to an underestimation of the genetic structure of the species accompanied with outlandish claims about continent-wide migration rates. Yet, when whole genome data was finally used to tackle questions of phylogeography, a tremendous amount of fine-grained population structure was revealed in Europe and North America (see Nunez 2020, and 2021). These studies have shown that much of population genetics before the onset of NGS had a “power” problem. I am afraid this paper has also fallen victim of the same problem. Now, for B. glandula, the power issue is particularly pronounced for two reasons: first, the combination of large Ne and large (but not infinite!) migration rates often results in complex levels of population structure such that large numbers of both n and m are needed to solve the Patterson equation. This is the reason why most modern demographic studies on wild systems use tools such as RAD, ddRAD, GBS, or Pool-Seq to sequence large amounts of genetic data in a cost-effective way. The second issue unique to B. glandula is the fact that, as a recent invader, it most definitely is not in any sort of demographic equilibrium. As such, many population genetic tests which assume some “approximation of equilibrium demography" will not be very useful here.

How to improve this shortcoming? The author could demonstrate, mathematically or using simulations that the data (in its current form) has sufficient power to address the question. Alternatively, they could sequence more genomic markers to ensure that the dataset has the adequate level of statistical power.

In line 154 the author makes a claim about temperature selection. I am very puzzled by this assertion. Given that barnacles have high Ne, their recombination blocks genome-wide are small, relative to, for example, humans. This genomic landscape of recombination all but guarantees that selection will be detectable only at the “local” genomic level. With this being said… the author only has at his disposal two markers… And both markers are often understood as strongly neutral, or more appropriately, strongly conserved by purifying selection. As such, these markers have no power whatsoever to substantiate claims about selection across the whole genome! What the author could say is that neither COX-I nor EF-1 are under selection… but given what we know about these genes as demographic markers, that seems a forgone, if not trivial, finding. Moreover, there is an inherent contradiction here: if the markers where indeed to be under selection, then all the demographic assertions would have been invalidated! I suppose that sampling these markers across the tidal range could be of demographic value if the author was attempting to ask a demographic question about priority settlement, but that would return us back to the issue of power.

How to improve this shortcoming? Tackling questions of selection requires appropriate genomic sampling. So, unless the author can add that to the data-set, concomitant with the appropriate analyses, any claims of selection are unsubstantiated. If this cannot be achieved, then the author must refrain from making these claims.

In line 160, the author claims to have presented "strong evidence" for Alaska as the ancestral population. I cannot find any formal test of the data which would substantiate this claim.

How to improve this shortcoming? Perform an appropriate demographic analysis where the ancestral-derived population history is explicitly tested. Such a test would require more than two linked markers.

Additional comments

Line 67: The correct citation is “Nunez” not “Nuez”

Line 104: please show the linear formulas used in the model fitting described.

Line 119: how long is CO-I and how long is EF-1 by themselves? This is unclear here

Line 121: The term “dominant” means something very explicit in genetics. I think the author means “most abundant”.

Line 169: The authors say that they did not “detected no genetic variation”… do they mean “no structuring of the genetic variation”

Figure 1: I am having a very hard time looking the bars in Figure 1B/C, please increase the spread of the points and error bars. You can do this with, for example, “position=position_dodge(width=0.5)” within each of the ggplot objects.

·

Basic reporting

I found the article easy to read and appropriately referenced.

Experimental design

I suggest the author justify exploration of the intertidal gradient and temperature gradient.

It is important to regress the Haplotype freqeuncies regressed against sea surface temperature or air temperature. In the native North American cline, there are shifts in haplotype frequencies in central California where it's relatively warmer. If the populations at the southern edge of the current distribution in Japan are not similar in temperature to central California, then we may not expect that a cline should occur in Japan.

Can you expand on why we should expect mid- to high-intertidal zones to differ in haplotype frequences? I am certain it's because of desiccation and temperaure shifts, but you should explain that more thoroughly. You should also address that there can be hotspots in which the mid-intertidal is more stressful at a high latitude than at low latitudes because of differences in the timing / dynamics of tidal cycles (Helmuth et al 2006 Ecological Monographs, 76, 461–479)

Validity of the findings

the findings are appropriate, but the author should address the limitations of the study design.

Additional comments

I wonder if there is more information about invasion genetics you can glean from analyses that ignore the three clades. For instance, are there any differences with latitude in nucleotide diversity or haplotypic diversity? Do you see an increase / decrease in nucleotide or haplotypic diversity between 2004 and 2019? What about between native and introduced populations?

minor comments
51: I might expand the phrase “global shipping” to include the possibilities of hitch-hiking on wood (Kado 2003), hull fouling and/or ballast water.
57: Geller concluded that they could not discern whether the source of Japanese was Puget Sound alone, or a mix of Puget Sound and populations from Puget Sound thru Alaska.
130: you may want to re-state the Haplotype frequencies from 2004 for readers.
142: it’s impossible to know if the cline was caused by thermal selection. It is more likely that the cline was generated by secondary contact of historically separated populations. There is evidence however, that the cline is maintained by selection (Sotka et al 2014, Wares et al 2021).
Figure 1: I only see one pie chart for MR and OF. are those high or low IT?s

---

## Round 0.2 · Minor Revisions

Thanks for the detailed comments and responses. There's 2 minor comments to address from Reviewer 1 (see line 170 and 191).

Reviewer 1 ·

Basic reporting

I had a concern regarding data availability and permits --> The authors have addressed my concerns successfully.

Experimental design

I had no major issues on this review area.

Validity of the findings

My original concern had to do with some of the claims of population structure given the limited scope of the genetic sampling --> The authors have included additional information in their discussion, and in the response to reviewers letter, which contextualizes their findings and address the bulk of my concerns. I have a minor comment about the writing of this claim, see below. Assuming my minor claim is addressed, I am satisfied with their response.

I also had a concern about some claims of temperature selection which have now been clarified in the text. I am satisfied with this change.

Additional comments

Note that lines numbers come from the PDF doc:

Line 53. it reads ".. this species was introduced to the Atlantic coasts of Argentina". Here, the meaning of "This species" is lost by the intervening sentence. I recommend starting a new sentence all together. Example: "By means of hitch-hiking on wood (Kado 2003), hull fouling, and/or ballast water. B. glandula was introduced to the Atlantic coasts of Argentina ..."

63: the word "previously" is implied so you don't need it.

104: I am a bit confused by the naming conventions of the haplotypes. Both COI and EF1 have A, B, C haplo-classes? a bit more clarity is recommended.

170: the phrase "whereas these DNA markers detected a clear latitudinal genetic cline in the native range (Sotka et al. 2004)." is very confusing. The crux of the paper is that Sotka et al 2004 had found a cline, where as the current paper does not find it. Given that is a core finding of the paper, this sentence must be said explicitly. Please re-write that sentence and say it clearly.

191: "I detected no structuring of genetic variation in the introduced barnacle" this claim has to be toned down a bit ... The author has to clarify that this lack of genetic structure is seen at the studied markers, and it is inferred to be a proxy for the global pop. structure of the species (which is fine, but it must be stated).

---

## Round 0.3 · accepted · Accept

Thanks for addressing the final points by the reviewer. I'm happy to accept this.